**PLOS** NEGLECTED TROPICAL DISEASES

# Clinical sensitivity and time-to-result of a cascaded pooled testing approach for assessing the prevalence and intensity of *Schistosoma haematobium* infection

Abraham Degarege[1]*, Bruno Levecke[2], Yohannes Negash[3], Abebe Animut[3], Berhanu Erko[3]

1 Department of Epidemiology, College of Public Health, University of Nebraska Medical Center, Omaha, Nebraska, United States of America, 2 Department of Translational Physiology, Infectiology and Public Health, Faculty of Veterinary Medicine, Ghent University, Merelbeke, Belgium, 3 Aklilu Lemma Institute of Pathobiology, Addis Ababa University, Addis Ababa, Ethiopia

* abraham.mengist@unmc.edu

**Data Availability Statement:** All data underlying findings in this submission is fully available, without restriction as supplementary files.

## Abstract

### Background

This study compared the clinical sensitivity and the time-to-result of an individual testing (IT) and a cascaded pooled testing approach (CPT; a positive test result in a pooled sample triggers examination of smaller-sized pools or individual samples) for assessing the prevalence and the intensity of *Schistosoma haematobium* infection. We also compared the sensitivity of the CPT in detecting *S. haematobium* infection when deploying urine filtration microscopy (UFM) *vs.* urine reagent strips (URS), and testing 10 mL *vs.* 15 mL of urine.

### Methodology/Principal findings

Between October 2021 and April 2022, *S. haematobium* eggs were counted in urine samples collected from school-aged children living in the Afar and Gambella Regional States of Ethiopia. Urine samples were collected at baseline (n = 1,288), and one month after administration of praziquantel (n = 118). All urine samples were processed through both an IT and a CPT approach (pools of 5, 10, 20, and 40 individual samples), deploying UFM (10 mL) and URS (10 mL). In addition, 15 mL urine was processed through the CPT deploying UFM. At baseline, the prevalence of *S. haematobium* infection estimated when using UFM and deploying a CPT approach was significantly lower (17.3%) compared to an IT approach (31.5%). The clinical sensitivity of the CPT in detecting *S. haematobium* eggs was 51.7%. The sensitivity increased significantly as a function of increasing log transformed urine egg counts (UECs) of the individual samples (OR 2.71, 95%CI 1.63 — 4.52). The sensitivity was comparable when the amount of urine examined was 10 mL (51.7%) *vs.* 15 ml (50.8%), and when UFM was used for testing *vs.* URS (51.5%). The mean log UECs estimated following the CPT approach was lower compared to the estimate by the IT (*p* <0.001). UECs of the individual samples estimated using the IT and CPT approaches were moderately correlated

**Funding:** This work was supported by the College of Public Health at the University of Nebraska Medical Center through the college Innovation Fund Program for collaborative and interdisciplinary public health projects to AD, BL, AA and BE. The funder had no role in study design, data collection and analysis, decision to publish, or preparation of the manuscript.

**Competing interests:** The authors have declared that no competing interests exist.

(r = 0.59 when 10 mL and 15 mL urine was examined after pooling). CPT reduced the time needed for processing urine samples and testing for *S. haematobium* infection by 29% with UFM and by 27.7% with URS.

## Conclusions/Significance

CPT based on UFM and URS techniques may help to rapidly identify areas with higher prevalence of *S. haematobium* infection (hotspots) in a population. However, the performance of this approach in estimating the prevalence of infection may be compromised, particularly in endemic areas with low intensity infection.

## Author summary

We examined the sensitivity and the time-to-result of a cascaded pooled testing approach for detecting and estimating the intensity of *Schistosoma haematobium* infection. Urine samples collected from school-aged children in the Afar and Gambella regions of Ethiopia were analyzed individually and in pools using both urine filtration microcopy (UFM) and urine reagent strips (URS). The cascaded pooled testing approach detected close to 52% of urine samples with *S. haematobium* eggs. The sensitivity increased as the egg count of the individual samples increased but was comparable when the amount of urine examined was 10 mL *vs.* 15 mL, and when the diagnostic test used was UFM vs. URS. Pooled testing reduced the time needed for processing urine samples and counting *S. haematobium* eggs by 29%. Pooled testing based on UFM may help rapidly identify areas with higher prevalence of *S. haematobium* infection in a population.

## Background

Neglected tropical diseases (NTDs) present an important public health burden, hindering progress toward sustainable development goals in endemic countries [1]. Out of the 20 diseases formally recognized as NTDs by WHO, schistosomiasis, which is caused by infections with flatworms of the genus *Schistosoma*, contributes to more than 11% of the overall burden attributable to NTDs [2]. The most important *Schistosoma* species that affect humans are *S. haematobium*, *S. japonicum*, and *S. mansoni*. They infect more than 250 million people worldwide and are particularly prevalent in sub-Saharan Africa, causing either urogenital schistosomiasis (*S. haematobium* only) or intestinal schistosomiasis (all other species) [3–5]. Urogenital schistosomiasis is associated with malnutrition, cognitive deficiency, and bladder cancer, while intestinal schistosomiasis causes abdominal pain, diarrhea, weight loss and anemia [3–5].

To reduce the schistosomiasis attributable morbidity, praziquantel (single oral dose of 40 mg/kg body weight) is periodically administered to school-aged children (SAC) living in endemic regions [6–8]. To initiate and subsequently monitor the progress of these large-scale deworming programs, periodically assessing the prevalence in nationwide surveys is warranted to down- or upscale the frequency of deworming whenever necessary [6–8]. As of now, the data on the prevalence and intensity of *S. haematobium* infection required for the initiation or evaluation of mass treatment programs is typically gathered by individually screening subjects using urine filtration microscopy (UFM) in large-scale epidemiological surveys. However, these surveys are resource demanding, and more cost-effective strategies are needed.

A cost-saving alternative is a pooled sample testing approach [9–13]. In 2015, we provided a proof-of-principle of pooled sample testing approach to assess both the occurrence and the intensity (using the number of eggs in 10 mL urine as a proxy) of *S. haematobium* infection in a population in Afar, Ethiopia [14]. However, this study had some important shortcomings. First, we deployed UFM only, which is known to be less sensitive in detecting light *S. haematobium* infection [5]. Other methods that involve the detection of *S. haematobium* DNA (e.g., PCR, loop-mediated isothermal amplification (LAMP)), antigen (e.g., point-of-care circulating cathodic antigen (POC-CCA)), or antibody released by the body against the parasite (e.g., ELISA, Indirect Hemagglutination Assay (IHA)) are available to diagnose *S. haematobium* infection [5]. Each method has issues with accuracy, efficiency, or accessibility [5]. Alternative cheap point-of-care tests are urine reagent strips (URS). Second, we did not fully explore the application of pooling samples. We reported only the sensitivity of pooled examination strategy in detecting infection at the group level (with sobering results), but we never explored the potential of a cascaded pooled testing (CPT; a positive test result in a pooled sample triggers examination of smaller-sized pools or individual samples) approach to determine the prevalence of infection at the population level. Finally, it remains unclear whether such a CPT would reduce the time-to-result, and therefore can be recommended as a cost-efficient alternative to the current diagnostic approach. In the present study, we compared the clinical sensitivity and the time-to-result of an individual testing (IT) and CPT approach for assessing the prevalence and intensity of *S. haematobium* infection deploying both UFM and URS.

## Methods

### Ethics statement

The study protocol was approved by the institutional review boards of the University of Nebraska Medical Center (USA; Ref No. IRB#908-19-EP) and Aklilu Lemma Institute of Pathobiology, Addis Ababa University (Ethiopia; Ref. No. ALIPB-IRB/10/2012/20). The District Health Offices, school authorities, and teachers were informed about both the purpose and the procedures of the study. Only those children who agreed to participate and whose parents or guardians gave verbal informed consent were recruited for this study. Children who were infected with *S. haematobium* based on either UFM or tested positive on URS were treated free of charge with a single oral dose of praziquantel (40 mg/kg body weight).

### Study design

Between October 2021 and April 2022, we visited six villages in Gambella (Village-17 in the Abobo district) and Afar Regional States of Ethiopia (Andada, Burri, Hasoba, Kelhat and Kusra) [15]. *S. haematobium* is endemic in these villages in Gambella and Afar Regions, with prevalence ranging from 24.2% to 35.9% in Gambella and 5.3% to 37.0% in Afar school children [16–18]. On arrival, the District Health Office, village administrators and school directors were informed about the purpose and the procedures of the study, who subsequently informed the community about the study. Then, children were given labeled plastic containers and demography data including age, gender and villages were collected when they brought the urine specimens. In total, 1,288 urine samples (+/- 80 mL) were collected from an equal number of SAC. SAC were provided a single oral dose of praziquantel (40 mg/kg body weight) when *S. haematobium* eggs were found via the UFM or when URS tested positive. One month later, SAC that tested positive using either diagnostic method were asked to provide a second urine sample. It was anticipated that following-up these children would provide additional samples with low urine egg count (UECs).

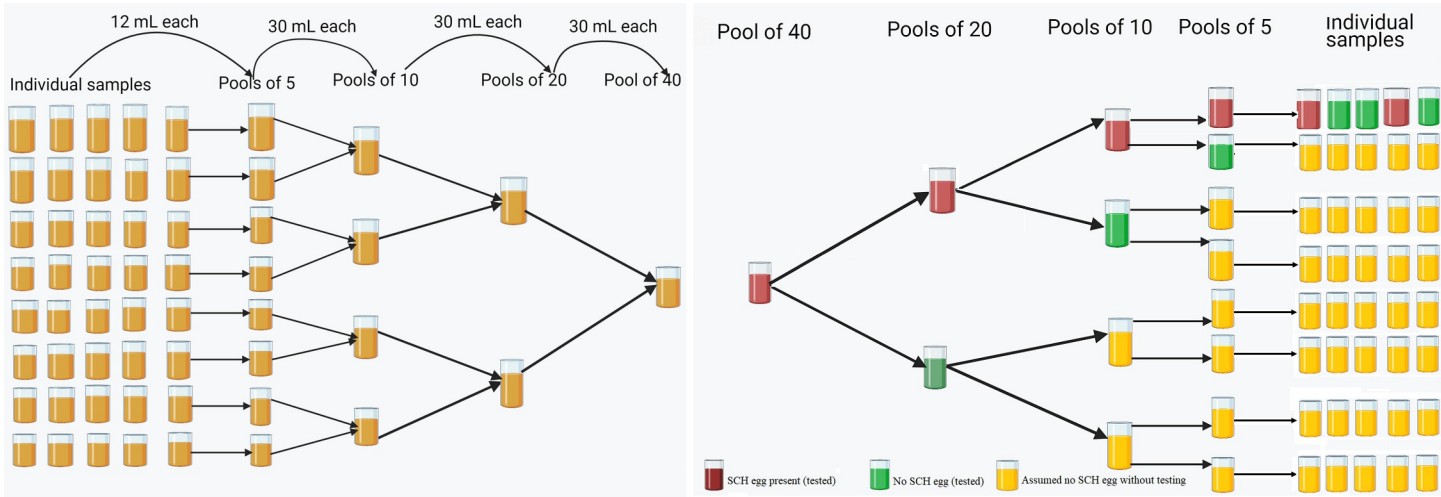

**Fig 1. Cascaded pool testing strategy.** This figure illustrates the procedure to obtain pools of 5, 10, 20 and 40 individuals (**Panel A**), and the cascaded pooled sample testing approach to analyze pools of 5, 10, 20 and 40 individual samples (**Panel B**). Fig created using biorender (https://www.biorender.com/).

In the field, 10 mL of each individual urine sample collected at baseline and one month after praziquantel treatment was screened with both UFM and URS. The remaining 70 mL urine sample was transferred to a vial containing 0.7 mL formalin (37% formaldehyde) and transported to the Medical Parasitology Laboratory of Aklilu Lemma Institute of Pathobiology, Addis Ababa University. The UFM involved filtering of urine samples through polycarbonate filter membranes (13 mm diameter and 12 to 20 μm pore size) using a syringe, followed by examining the filters under a microscope to quantify *S. haematobium* eggs [19]. The URS method entailed detecting microscopic blood (microhematuria) in urine samples, which served as a proxy for *S. haematobium* infection [19]. This was accomplished using Combur 10 Test reagent strips following the manufacturer's instructions.

In the laboratory, individual urine samples were first pooled following a previously described methodology (**Fig 1A**), which consisted of four consecutive steps. In the first step, we determined the number of pools of 40 samples to be made at the village (when the sample size was enough) or region level. Then, 40 plastic vials containing urine of 40 individuals were arranged in increasing order of their unique identifier number (ID) in 8 rows/groups of 5 individual samples. Assuming no link exists between individual ID and egg count, we considered the samples to be arranged randomly. Following that, pools of 5 individual samples were prepared by transferring 12 mL urine from each vial of the same row to a 60 mL vial. In a third step, pools of 10 individual samples were prepared by transferring (after shaking) 30 mL urine from the two different vials containing pooled samples of 5 individuals to a 60 mL vial. In a fourth step, pools of 20 individual samples were prepared by transferring (after shaking) 30 mL urine from the two different vials containing pooled samples of 10 individuals to a 60 mL vial. In a fifth step, pools of 40 individual samples were prepared by transferring (after shaking) 30 mL urine from two different vials containing pooled samples of 20 individuals to a 60 mL vial. Pools were created using samples from the same village or region without regard for the individual villages from which the samples were collected. Urine sample collection, examination, and pooling procedures were similar at baseline and one month after praziquantel treatment.

After pooling the samples, each pool of 40 individual urine samples were processed by applying a CPT approach. Generally, in case a pool of 40 tested positive, the 2 corresponding pools of 20 individual samples were processed. Here, too, when any pool of 20 tested positive,

each corresponding pool of 10 was tested separately. This process was repeated until any of the pools were revealed to be negative or when all individual samples were analyzed. **Fig 1B** further illustrates the CPT approach. To evaluate the performance based on the volume of urine examined and diagnostic test, CPT approach was performed by deploying UFM on 10 mL and 15 mL urine samples and URS.

To verify whether our CPT saves time for testing, we recorded the time required (i) to make pools of 5, 10, 20, and 40 individual samples; (ii) to process samples through an IT or CPT approach deploying UFM only, and (iii) to count *S. haematobium* eggs in either an individual or a pooled urine sample applying UFM. As the time spent processing samples and checking *S. haematobium* eggs using URS was not collected during baseline data collection, time data for the IT and CPT using URS was estimated retrospectively. Testing for *S. haematobium* infection using the URS involves dipping the strip in urine, waiting for a period, and then reading the results [20]. This process takes approximately 2 minutes. All samples were examined by experienced laboratory technicians, decreasing the chances of missing low intensity *S. haematobium* infection.

## Data collection form and management process

In this study, a carefully designed form was used to ensure the collection of accurate data points directly relevant to the study's objectives including demography, which consists of age in years, gender, region and village names where the children live, and lab data including *S. haematobium* egg counted through UFM technique and results about the presence of blood in urine through URS. The form was structured to provide unambiguous questions and collect quantitative and qualitative data. To ensure consistency and easier comparison and aggregation of data across different villages and regions, standardized gradings/units were used to record the test results of URS and UFM. The URS test results were documented as zero (negative), ± (trace), + (weak), ++ (moderate), and +++ (strong) erythrocytes per microliter of urine. The UFM results were recorded as the number of *S. haematobium* eggs counted in 10 ml (IT and CPT approaches) or 15 ml urine (CPT approach).

Robust data management procedures were implemented to ensure the integrity, confidentiality, and accessibility of the data. In the field, data were collected on paper forms and manually entered into a secure electronic database after the lab examination was completed. Once one person entered the data, a second person cross-checked it, and discrepancies were resolved through discussion. After removing personally identifiable information, all data were stored in a secure, password-protected local hard drive, with access restricted to authorized personnel only. Manual reviews were conducted before analysis to identify and correct inconsistencies, missing values, and outliers to ensure data quality. The data were deposited in reputable data repositories for long-term preservation and accessibility. Metadata, including descriptions of variables and coding schemes, was provided to the PLOS NTD journal to enable other researchers to understand and utilize the data effectively and reproduce the results.

## Sample size

The prevalence of *S. haematobium* infection was 20% in Afar and 35.9% in Gambella Regional States of Ethiopia [17,18]. A previous study showed that UFM based pooled testing (based on pools of 10 samples) is 68.6% sensitive to determine the prevalence of *S. haematobium* infection in a population [14]. Hence, to accurately estimate the sensitivity of the pooled testing in assessing the prevalence of *S. haematobium* infection with 80% power and 95% CI, we need a minimum sample of 708 in the Afar and 410 in the Gambella [21].

## Statistical data analysis

The prevalence of infection at baseline was determined by dividing the number of individual urine samples tested positive for *S. haematobium* eggs deploying UFM or URS by an IT or CPT approach by the total number of individual samples tested. Chi-square test was used to compare the prevalence of infection by age groups (5–9 years *vs*. 10–15 years), gender (male *vs*. female) and village where the children live. The mean intensity of infection at baseline was determined by averaging the number of egg count per 10 mL urine samples using the IT and CPT approach. As the distribution of the egg count was not normal even after cubic, square, square root, or log transformation, two-sample Wilcoxon rank-sum (Mann-Whitney) test was used to compare mean egg count by age groups, gender and the region where children live, and the Kruskal-Wallis test was used to compare mean egg count by villages of the children. To further explore sources of variation in the prevalence and intensity (UEC) of *S. haematobium* infection, we built Generalized Linear Models (GLM) with the test result (positive vs. negative and log urine egg count, UEC) as outcome variables. We used age groups (5–9 years *vs*. 10–15 years), gender (females vs. males) and villages (6 villages) as fixed effects and the region where children live as a random variable.

To determine the clinical sensitivity of either IT and CPT approaches based on UFM and URS, we used the combined test results of the IT and CPT approaches based on the UFM technique as reference method. An individual sample was defined as 'true' positive when at least one egg was found in an individual sample following either an IT and/or a CPT approach. In other words, an individual sample was considered positive when IT tested positive and/or CPT tested positive (i.e., IT+ and/or CPT+ was considered positive), and an individual sample was considered negative when an IT and CPT tested negative (i.e., IT- and CPT- was considered negative). By doing so, the specificity of both testing approaches was assumed to be 100% due to the morphology of the eggs. A chi-square for trend analysis was conducted to examine the linear trend of sensitivity of the CPT with an increase in pool sizes from 5, 10, 20 to 40.

To gain insights into the impact of infection intensity on the clinical sensitivity of a pooled examination, we assessed the change in sensitivity as a function of the mean UEC across the individual samples that made the pools. To this end, we classified the intensity into four levels. The levels were based on the quartiles (Q) of the mean UECs of individual samples across all pools of 5, 10, 20 and 40 that were tested deploying UFM (level 1: mean UECs $\leq$ Q1; level 2: Q1 < mean UECs $\leq$ Q2; level 3: Q2< mean UECs $\leq$ Q3; level 4: mean UECs > Q3). Then, we built a GLM with the pooled test result (positive/negative) as an outcome and the log of the mean UECs of the corresponding individual urine samples, volume of urine and the interaction term as covariates. The region where the study participants live was included as a random variable. As URS was deployed to examine 10 mL of urine only, the test results based on the examination of pools with URS were not included in the GLM. To verify the ability of a CPT approach to determine infection intensities when deploying UFM, we assessed the linear relationship in UECs estimated by the CPT and IT using Pearson correlation ('r') test [22]. The difference in the mean log UECs was verified using a paired t-test.

The percent time saved in counting *S. haematobium* eggs due to CPT approach was estimated by calculating the difference in time spent for counting *S. haematobium* eggs deploying the CPT and IT approaches. The total time to count *S. haematobium* eggs for the IT approach was the sum of the time for preparing (i.e., time for placing filters in the filter holder and filtering the sample) and examining individual urine samples. The total time to count *S. haematobium* eggs for the CPT strategy included the time for pooling besides the time for preparing (i.e., time for placing filters in the filter holder and filtering the sample) and examining pooled and individual urine samples. The level of significance used was $p < 0.05$.

## Results

### Prevalence and intensity of *Schistosoma haematobium* infection among the study participants

A total of 1,288 SAC were included in the study, of whom 118 also provided a urine sample one month after being treated with praziquantel, resulting in a total of 1,406 individual urine samples. Among the 1,288 children, 58.3% were boys and 51.6% were between 5 to 9 years old. Table 1 summarizes the baseline prevalence and the intensity of *S. haematobium* infection based on combined IT and CPT approaches deploying UFM across the different age groups, sexes and villages. At baseline, the prevalence of *S. haematobium* infection was 33.5% (431/1,288). Infection was more prevalent in boys (34% *vs.* 32.6%), older children (age 10 to 15 years: 37.6% *vs.* age 5 to 9 years: 29.5%), and highest in Village-17 within the Abobo district of Gambella Region (43.8% *vs.* 37.8% (Buri) *vs.* 28.9% (Kusra) *vs.* 20.6% (Kelhat) *vs.* 10.8% (Andade) *vs.* 2.3% (Hassoba)). The mean UECs was significantly higher in males than females ($p = 0.002$) (Table 1).

In the GLM that included age group, sex and villages as fixed effects and region as a random variable, the odds of *S. haematobium* infection was significantly lower in children who live in villages of the Afar region (Kusra aOR: 0.54, 95% CI: 0.39–0.76; Hassoba aOR: 0.03, 95% CI: 0.010–0.099; Kelhat aOR: 0.33 95% CI: 0.14–0.77; Andada aOR: 0.16; 95% CI: 0.05–0.45) compared to those who live in Village-17 of the Abobo district in Gambella region. The log UECs were also significantly lower in children who live in villages of the Afar region (Kusra β: -0.46, 95% CI: -0.74—-0.18; Hassoba β: -4.94, 95% CI: -7.30—-2.57; Kelhat β: -1.19 95% CI: -2.01 —-0.37; Andada β: -1.52; 95% CI: -2.41—-0.63) than those in Village-17 of the Abobo district in Gambella region. The log UEC was greater in males than females (β: 0.30 95% CI: 0.09–0.52). However, the differences in the prevalence and intensity (log UEC) of infection was not significant between males and females.

Table 1. Prevalence and intensity of *Schistosoma haematobium* infection among school age children in Afar and Gambella regional states of Ethiopia, October 2021 to April 2022.

| Variables | Category | Number Examined | Perentage IT/CPT positive smples by UFM (95% LCL—UCL) | P-value | Arithmetic mean egg count (geometric mean) by UFM | P-value | Percentage IT/CPT positive samples by URS (95% LCL—UCL) | P-value |
|---|---|---|---|---|---|---|---|---|
| Age in | 5 to 9 | 665 | 29.5 (26.0–33.0) | 0.002 | 1.66 (1.50) | 0.742 | 33.1 (29.5–36.7) | 0.100 |
| years | 10 to 15 | 623 | 37.6 (33.8–41.4) | | 1.78 (1.53) | | 37.5 (33.6–41.3) | |
| Gender | Female | 538 | 32.7 (28.7–36.7) | 0.613 | 1.49 (1.39) | 0.002 | 33.2 (29.2–37.2) | 0.201 |
| | Male | 750 | 34.0 (30.6–37.4) | | 1.88 (1.67) | | 36.7 (33.2–40.1) | |
| Villages | Buri | 352 | 37.8 (32.7–42.8) | | 1.83 (1.72) | | 42.6 (37.4–47.8) | |
| | Kusra | 256 | 28.9 (23.4–34.4) | | 1.67 (1.52) | | 35.9 (30.0–41.8) | |
| | Hassoba | 131 | 2.3 (-0.3–5.0) | <0.001 | 0.23 (0.69) | 0.069 | 4.5 (0.09–8.2) | <0.001 |
| | Kelhat | 34 | 20.6 (7.0–34.2) | | 1.15 (1.06) | | 20.6 (6.3–34.9) | |
| | Andada | 37 | 10.8 (0.7–0.21) | | 1.58 (1.30) | | 13.5 (2.0–25.1) | |
| | Village-17 | 478 | 43.8 (39.4–48.0) | | 1.72 (1.50) | | 40.5 (36.0–44.9) | |
| Regions | Afar | 810 | 27.3 (24.2–30.4) | <0.001 | 1.73 (1.60) | 0.217 | 32.1 (28.9–35.3) | 0.001 |
| | Gambella | 478 | 43.8 (39.4–48.2) | | 1.72 (1.50) | | 40.8 (36.5–45.2) | |

Individuals with IT and/or CPT positive results by UFM were used as denominators when calculating the mean egg count per 10 mL urine.

IT: individual testing. CPT: cascaded pooled testing. UFM: urine filtration microscopy. URS: urine reagent strips.

## Comparison of an individual and cascaded pooled testing strategy

**Clinical sensitivity.**   Overall, the clinical sensitivity of an IT approach across the 431 individual samples with *S. haematobium* eggs based on IT and CPT was 94.2% (95%CI: 91.6–96.2) when deploying UFM and 70.1% (95%CI: 65.5–74.3) when using URS. Compared to an IT approach, the clinical sensitivity was significantly generally lower for a CPT approach for both UFM (51.7%, 95%CI: 46.9–56.5) and URS (51.5%, 95%CI, 46.7–56.3) (**Table 2**).

The clinical sensitivity of the CPT in detecting *S. haematobium* eggs using UFM in individual urine samples increased significantly with an increase in the log mean UECs of the individual samples in the GLM that adjusted for the effect of volume of urine examined and accounted for the region where the samples were collected as a random variable (OR: 2.71, 95%CI 1.63–4.52). The highest clinical sensitivity of the CPT in detecting *S. haematobium* eggs in pooled urine samples was seen when the mean log egg count of the individual samples making the pools was greater than the third level quartile ranges (**Fig 2**). However, the sensitivity of CPT in detecting *S. haematobium* eggs in individual urine samples was comparable when the amount of urine examined was 10 mL (51.7%) vs. 15 mL (50.8%), and when UFM was used for testing *vs*. URS (51.5%). UFM was 100% specific in identifying urine samples without *S. haematobium* egg through the IT and CPT approach. However, the specificity of the IT and CPT approaches in declaring urine samples without *S. haematobium* egg was 86.9% and 91.1%, respectively when URS was employed for testing.

The proportion of urine samples tested positive for *S. haematobium* egg using UFM, which was truly positive, was 100% through the IT and CPT approach when the urine volume examined was 10 ml. The proportion of urine samples tested negative for *S. haematobium* egg using UFM that were true negative was 97.2% through the IT; 80.5% by the CPT when the urine volume examined was 10 ml and 80.2% when the urine volume examined was 15 ml. However, the positive predictive values of URS were 73.0% by the IT and 74.5% by the CPT. The negative predictive values of URS were 85.2% by the IT and 78.9% by the CPT.

Table 3 summarizes the sensitivity of the cascaded pooled testing in detecting *S. haematobium* eggs in pooled urine samples using UFM and URS techniques. All 34 pools of 40 samples were tested, and 24 (72.7%) were found positive when 10 ml urine was examined using UFM. As a result, the corresponding 48 pools of 20 samples (out of 64 pools of 20 samples positive with *S. haematobium* egg and/or with non-zero mean UEC of individual samples), which made the 24 positive pools of 40 samples, were tested, and 38 (59.4%) were found positive when 10 ml urine was examined using UFM. Consequently, 76 pools of 10 samples (out of 129 pools of 10 samples positive with *S. haematobium* egg and/or with non-zero mean UEC of individual samples), representing the 38 positive pools of 20 samples, were examined, with 75 (58.1%) testing positive when 10 ml urine was examined using UFM. Then, the 150 (out of 223

**Table 2. Sensitivity of the individual and cascaded pooled testing strategy in detecting *Schistosoma haematobium* infection by the diagnostic method and the volume of urine examined.**

| Diagnostic test | Testing strategy (Volume of urine examined) | | | |
|---|---|---|---|---|
| | Individual testing (10 mL) | Cascaded pooled testing (10 mL) | Cascaded pooled testing (15 mL) | Difference between IT and CPT when using 10 ml samples (95% LCL—UCL) |
| UFM | 406/431 (94.2%) (91.6–96.2) | 223/431 (51.7%) (46.9–56.5) | 219/431 (50.8%) (46.0–55.6) | 42.5% (37.3–47.7)*** |
| Urine reagent strips | 302/431 (70.1%) (65.5–74.3) | 222/431 (51.5%) (46.7–56.3) | - | 18.6% (12.2–25.0)*** |

Note: the number of positive samples based on combined IT and CPT approaches deploying UFM used as a reference to calculate the sensitivity values (n = 461).

***: *P* value <0.001

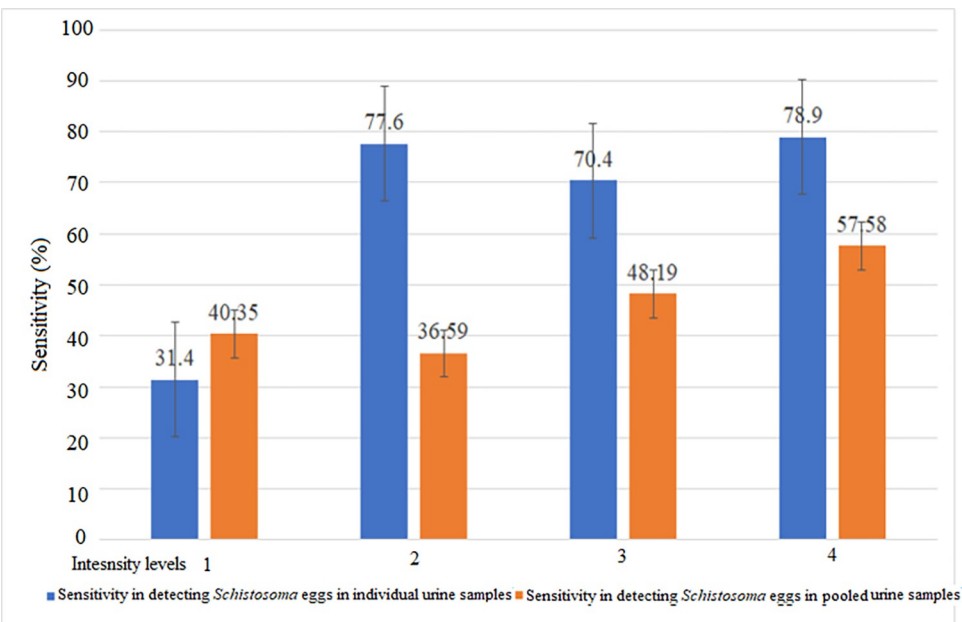

**Fig 2. Sensitivity of detecting *Schistosoma haemtobium* eggs deploying an individual and pooled cascaded testing approach stratified by the quartile ranges of the egg count from the individual samples deploying the individual testing strategy.** The lines on the top of each bar represent the differences in the upper and lower 95% confidence interval. Intensity levels 1: mean UECs ≤ Q1 (25th percentile); level 2: Q1 < mean UECs ≤ Q2 (50th percentile); level 3: Q2< mean UECs ≤ Q3 (75th percentile); level 4: mean UECs > Q3.

pools of 5 samples positive with *S. haematobium* egg and/or with non-zero mean UEC of individual samples) pools of 5 samples corresponding to the 75 positive pools of 10 samples were tested. Out of these, 109 (48.9%) were found positive when 10 ml urine was examined using UFM.

Overall, the clinical sensitivity of the UFM based CPT in detecting *S. haematobium* eggs in pooled urine samples tends to increase with an increase in pool size (Chi$^2$ for trend = 7.12, $P = 0.008$ when 10 ml urine was examined using UFM; Chi$^2$ for trend = 5.65, $P = 0.017$ when 15 ml urine was examined using UFM). However, the sensitivity of CPT in detecting *S. haematobium* eggs in pooled urine samples was comparable when results were compared based on diagnostic test (UFM *vs.* URS) or the volume of urine examined using UFM (10 mL *vs.* 15 mL) (Table 3).

**Table 3. Sensitivity of the cascaded pooled testing in detecting *Schistosoma haematobium* eggs in pooled urine samples.**

| Pool size | Total number of pools | Number of pools positive with *S. haematobium* egg and/or with non-zero mean UEC of individual samples | Number of pools checked for *S. haematobium* (remaining pools assumed negative for eggs without testing) | Number of pools with *S. haematobium* egg upon testing (sensitivity) | | | Arithmetic log mean UEC of pooled samples per 10 mL urine |
|---|---|---|---|---|---|---|---|
| | | | | UFM 10 mL | UFM 15 mL | URS | |
| 40 | 34 | 33 | 34 | 24 (72.7) | 23 (69.7) | 21 (63.6) | 1.00 |
| 20 | 70 | 64 | 48 | 38 (59.4) | 36 (56.3) | 40 (62.5) | 0.76 |
| 10 | 142 | 129 | 76 | 75 (58.1) | 72 (55.8) | 76 (58.9) | 0.75 |
| 5 | 283 | 223 | 150 | 109 (48.9) | 106 (47.5) | 125 (56.1) | 0.63 |

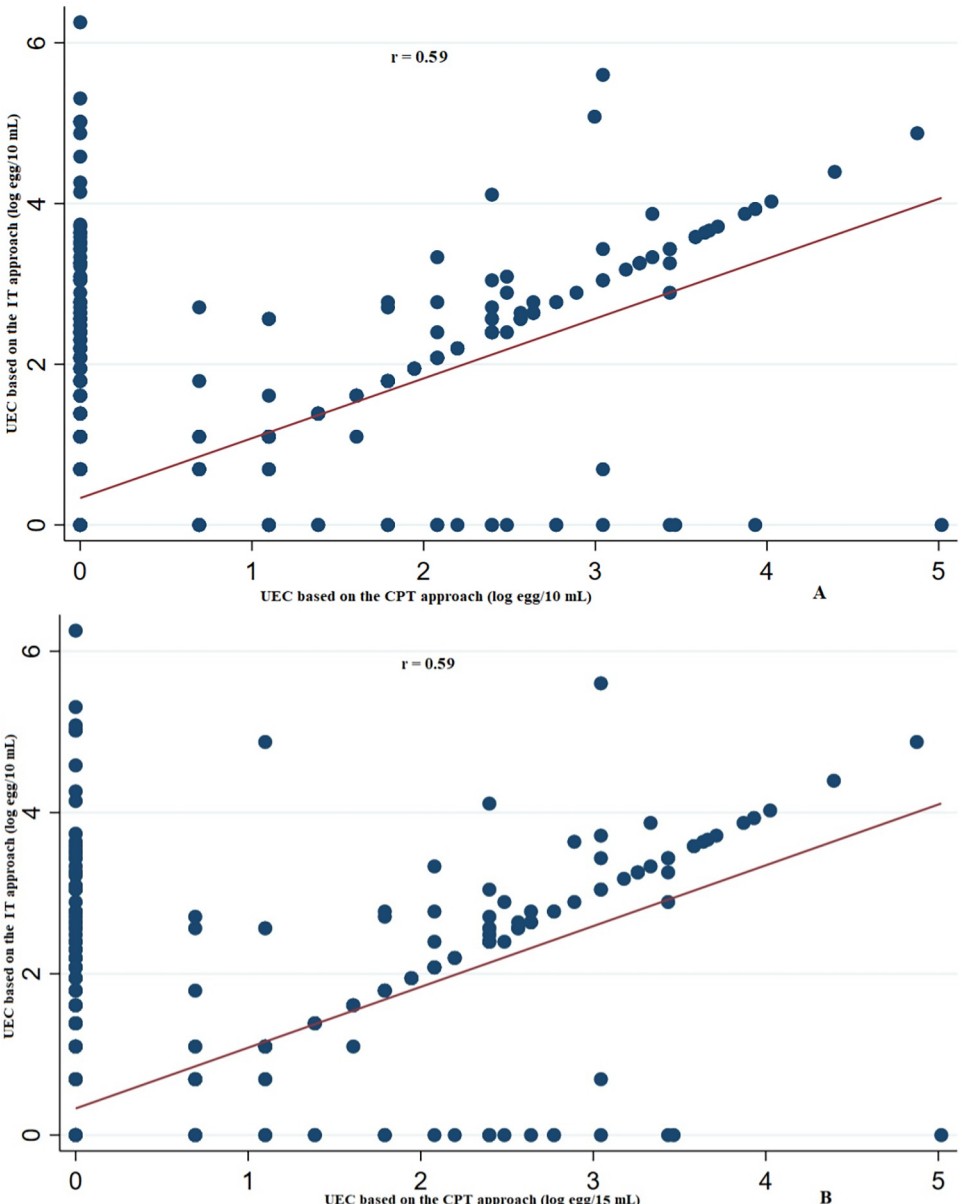

**Fig 3. The agreement in urine egg counts (UEC) estimated deploying the individual (IT) and the cascaded pooled testing (CPT) strategy (Panel A: 10 mL *vs.* Panel B: 15 mL).** The magnitude of correlation for each plot is based on the spearman correlation coefficient.

**Urine egg counts.** The arithmetic mean log UECs of the individual samples estimated following the IT approach (mean = 0.58 log egg /10mL) was significantly greater compared to the estimate based on the CPT (mean = 0.32 log egg/10mL, $p<0.001$). There was also a moderate correlation between the UECs of individual samples estimated following the IT approach and the CPT approach (r = 0.59) (**Fig 3**). The IT approach detected significantly more individual samples with *S. haematobium* egg which were missed by the CPT approach (points spread along either the axis).

**Time-to-result.** A total of 72.5 hours was required to filter and count *S. haematobium* eggs in 1,406 urine samples using the IT approach deploying UFM. However, 51.6 hours was

enough for the CPT approach to pool, filter 10 mL and count *S. haematobium* eggs in the 1,406 samples deploying UFM. CPT reduced the time needed for processing urine samples and counting *S. haematobium* eggs by 29%. Testing for *S. haematobium* infection using the URS takes approximately 2 minutes. A total of 46.9 hrs was needed to check *S. haematobium* infection from 1,406 urine samples through the IT approach by URS, but the time needed decreased to 33.9 hrs (27.7%) when the CPT was applied.

## Discussion

Urine filtration microscopy (UFM) based cascaded pooled testing approach (CPT) detected 50.3% of the individual urine samples with *S. haematobium* eggs. The sensitivity in detecting infection at the individual and pooled levels increased significantly with the increase in the number of eggs in the individual urine samples. There was a moderate correlation between UEC estimates using the IT and CPT approaches. The CPT approach significantly reduced the time required to diagnose *S. haematobium* infection.

The increase in the sensitivity of CPT with an increase in UEC is in line with our previous findings [14]. The dilution effect in pooling the samples and the low sensitivity of the UFM technique for detecting light intensity infection could have contributed to the decrease in the sensitivity of the CPT with a decrease in the intensity of infection [5]. The effect of dilution on sensitivity occurs because the concentration of *S. haematobium* eggs in a positive urine sample decreases as negative urine samples are added to the pool. Increasing the volume of urine (20 mL or 30 mL) and using more sensitive diagnostic methods (e.g., molecular tests, antigen detection methods) may improve the sensitivity of the CPT in detecting low intensity *S. haematobium* infection.

Prevalence data is key to decide on the initiation and scaling down or up of mass treatment programs [7]. In the current study, the prevalence of *S. haematobium* infection among the SAC when estimated using the CPT approach (17.4%) was lower compared to the estimate based on the IT strategy (31.3%). However, a significant number of urine samples of children identified as positive for *S. haematobium* egg by CPT were initially negative by the IT. This suggests that pooled testing may give biased estimates of the prevalence of *S. haematobium* infection at the individual level [23,24]. CPT might be used for large scale *S. haematobium* epidemiological surveys to rapidly identify hotspots for *S. haematobium* transmission, particularly where the intensity of infection is high. However, the method may increase the probability of false negative cases in areas where the intensity of infection is low. Hence, planning and evaluating the success of mass treatment programs based on CPT results should be made carefully. Further studies incorporating larger urine volumes, more sensitive diagnostic tests, repeated testing of the same samples or participants, optimal pool sizes, and varying levels of endemicity are necessary to provide recommendations on the utilization and timing of CPT for planning and assessing mass treatment programs against *S. haematobium* infection.

The scatter plot showed positive trends in the relationship of the mean UECs of the individual urine samples estimated based on the IT and the UECs estimated using the CPT (Fig 3). There was also a moderate correlation in UEC estimated by the IT and CPT approach. Previous studies also reported strong correlation in the average *S. haematobium* [14] or soil-transmitted helminths [25,26] egg count estimated from pooled and individual samples in humans [14]. These results strengthen the notion of applying pooled sample testing in large scale epidemiological survey to quantify the intensity of *S. haematobium* infection. However, the preferred pool sizes, volume of urine and diagnostic test that can best help estimate the intensity of infection at low costs across different endemicity levels need further investigation. Once the prevalence of *S. haematobium* infection and urine volume, along with the diagnostic test and

ranges of pool sizes that maximizes the sensitivity in detecting parasite eggs are identified, online tools can be utilized to determine the most effective pool size for efficiently screening the parasite at a low cost in large-scale surveys conducted in endemic regions [27].

The CPT approach significantly reduced the time needed to determine the status of *S. haematobium* infection in the study participants. Diagnosing a pool of 40 individual urine samples would allow testing of 1.5 times more samples instead of diagnosing individual samples. This allows screening of more subjects in different geographic regions within a relatively shorter time. Unlike the soil-transmitted helminths, *S. haematobium* has a focal distribution. Thus, pooled testing may help better map the diseases' distribution of high intensity infections by rapidly screening large number of populations. Previous studies also demonstrated a reduction in the time needed to pool, prepare, and examine Kato Katz thick smears for screening of soil-transmitted helminths infections through a pooled testing approach [13,26]. However, the percent reduction in time for testing due to pooling samples was lower [13] or greater [26] in the previous studies compared to the current one. Previous studies did not examine the individual samples making the pools when the corresponding pools were positive. On the other hand, the current study examined all the individual samples separately when the corresponding pool 5 samples were positive. In addition, unlike the previous studies, which applied fixed pool sizes (5 or 10), the current study applied four different pool sizes. This may suggest that the time reduction is not linearly related to the pool size. Moreover, various issues related to the experience of the laboratory staff examining the slides, the setup in the field or lab and the environment could contribute to the differences in the time needed for pooled testing.

In the current study, an IT approach that employed URS detected *S. haematobium* egg in urine samples at a greater sensitivity than the CPT approach which employed UFM. An IT based on URS may allow to diagnose *S. haematobium* infection faster than the CPT approach based on UFM. However, URS was less specific than UFM in declaring urine samples without *S. haematobium* egg in both the individual and pooled samples.

As far as we are aware, this study represents the first attempt to estimate the time required for pooling urine samples and to compare the sensitivity of UFM (utilizing volumes of 10 mL and 15 mL) with URS for diagnosing *S. haematobium* infection in pooled urine samples. However, this study had limitations. First, only UFM and URS were used for testing. Second, only testing 10 mL or 15 mL could have led to falsely declaring urine samples free of *S. haematobium* egg due to the dilution of pooled samples. One may consider examining more volumes of pooled urine samples or increase the number of urine samples examined per subject to reduce the dilution effect. As an illustration, in this scenario, it is anticipated that analyzing 3 x 10 mL of a pool containing 40 individual samples, rather than 10 mL, would elevate the sensitivity from 50.3% to 87.7% (= $1-(1-0.503)^3$). Obtaining a large volume of urine from children may not always be feasible and could impact the time required to obtain results. Offering water for drinking before urination could aid in collecting a sufficient volume of urine from children. Third, the sample size collected in some villages was insufficient to analyze the data at the village level with sufficient power. As the prevalence and intensity of *S. haematobium* showed variation by villages, performance of the pooled test may vary by the region where children live. Fourth, a hierarchical algorithm was followed to make the pools (pools of 40 made of pools of 20, pools of 20 out of pools of 10, pools of 10 out of pools of 5, and pools of 5 out of individual samples). The hierarchical algorithm helps to use urine wisely but may introduce bias as pools of 10, 20 and 40 are dependent on the pools of 5 and the individual samples contribute different volumes of urine to the pools. As the pool size increases, the contribution of individual urine samples to the pools diminishes proportionally. For instance, with a total pool volume of 60 ml maintained across different pool sizes, each individual sample contributes 12 mL to pools of 5, 6 mL to pools of 10, 3 mL to pools of 20, and 1.5 mL to pools of 40. Pooling

fixed volumes of individual samples separately for each pool size would help to reduce bias. Fifth, the sample size from the various villages was inadequate for conducting robust analyses for each area individually. Sixth, this study did not employ a highly sensitive independent gold standard test (e.g., PCR) as a reference to determine the sensitivity of CPT using UFM or URS. Finally, cost effectiveness of pooled testing was not evaluated as the cost utilized for collecting, processing and examining of the specimens was not collected. A cost-benefit analysis or simulation study would help to verify a cost-effective large-scale survey based on pooled testing for the control of *S. haematobium* infection [28]. Although pooling could lower cost by reducing the time for testing samples, the time needed to collect sufficient samples and resources needed for storing the samples could incur cost.

In conclusion, CPT based on UFM and URS techniques may help to rapidly identify areas with higher prevalence of *S. haematobium* infection (hotspots) in a population. However, the performance of this approach in estimating the prevalence of infection may be compromised particularly in endemic areas with low intensity infection. Further research or simulation studies that evaluates performance and cost of CPT across different levels of endemicity; number of samples collected and examined per individual; volume of urine examined, and sensitivity of the diagnostic technique is needed to decide how and when to apply pooled test for large-scale *S. haematobium* epidemiological surveys.

## Supporting information

**S1 Data. The prevalence and intensity of *Schistosoma haematobium* infection among school age children in Afar and Gambella regional states of Ethiopia, October 2021 to April 2022 deploying an individual and pooled cascaded testing approach using urine filtration microscopy and urine reagent strips.**
(XLSX)

**S2 Data. Urine egg counts (UEC) of *Schistosoma haematobium* infection estimated deploying the individual and the cascaded pooled testing strategy among school age children in Afar and Gambella regional states of Ethiopia, October 2021 to April 2022.**
(XLSX)

## Acknowledgments

We would like to thank the study participants for providing urine samples. We would like thank also the health professionals working at the health posts in the study areas, health offices in the districts and region levels and the community leaders for facilitating study participant recruitment and the data collection.

## Author Contributions

**Conceptualization:** Abraham Degarege, Bruno Levecke, Abebe Animut, Berhanu Erko.

**Data curation:** Abraham Degarege, Berhanu Erko.

**Formal analysis:** Abraham Degarege, Bruno Levecke.

**Funding acquisition:** Abraham Degarege, Bruno Levecke, Abebe Animut, Berhanu Erko.

**Investigation:** Abraham Degarege, Yohannes Negash, Abebe Animut, Berhanu Erko.

**Methodology:** Abraham Degarege, Bruno Levecke, Abebe Animut, Berhanu Erko.

**Project administration:** Abraham Degarege.

**Resources:** Abraham Degarege, Abebe Animut, Berhanu Erko.

**Software:** Abraham Degarege.

**Supervision:** Abraham Degarege, Yohannes Negash, Berhanu Erko.

**Validation:** Abraham Degarege.

**Visualization:** Abraham Degarege.

**Writing – original draft:** Abraham Degarege.

**Writing – review & editing:** Abraham Degarege, Bruno Levecke, Yohannes Negash, Abebe Animut, Berhanu Erko.

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
