## [Decision Letter · Decision Letter 0]

27 Feb 2024

Dear Dr Degarege,

Thank you very much for submitting your manuscript "Clinical sensitivity and time-to-result of a cascaded pooled testing approach for assessing the prevalence and intensity of Schistosoma haematobium infections" for consideration at PLOS Neglected Tropical Diseases. As with all papers reviewed by the journal, your manuscript was reviewed by members of the editorial board and by several independent reviewers. In light of the reviews (below this email), we would like to invite the resubmission of a significantly-revised version that takes into account the reviewers' comments. 

We cannot make any decision about publication until we have seen the revised manuscript and your response to the reviewers' comments. Your revised manuscript is also likely to be sent to reviewers for further evaluation.

Sincerely,

Daniela Fusco, PhD

Academic Editor

Francesca Tamarozzi

Section Editor

Reviewer's Responses to Questions

**Key Review Criteria Required for Acceptance?**

**Methods**

-Are the objectives of the study clearly articulated with a clear testable hypothesis stated?

-Is the study design appropriate to address the stated objectives?

-Is the population clearly described and appropriate for the hypothesis being tested?

-Is the sample size sufficient to ensure adequate power to address the hypothesis being tested?

-Were correct statistical analysis used to support conclusions?

-Are there concerns about ethical or regulatory requirements being met?

Reviewer #1: (No Response)

Reviewer #2: The objectives are clearly articulated and the study design is sound. The population used is appropriate and the sample size is sufficient and clearly outlined. Statistical approaches seem sound (though this is not my area of expertise) and there are no ethical concerns.

However, as described in the general comments further analysis is required. Most importantly this relates to the performance of URS, can it be considered the preferred option of the approaches tested? An analysis of specificity of URS in this cohort should be conducted, to be able to recommend such an approach.

Some additional post-hoc analysis should also be conducted regarding pool sizes. The authors currently have a varied number of pool sizes with existing data, I would encourage them to conduct a sub-analysis here – this may not be possible since as I understand smaller pool sizes were only tested in case of positivity, but I encourage the authors to revisit this data nonetheless. If this is not possible, I would encourage to at least have a detailed breakdown of the positive pools, as I believe this would still be informative. For example, starting at the largest positive pool size and describing within this pool how many sub-pools were positive, until the individual positive samples are identified. This is also true for the egg counts found in each pool. These data can be included as supplementary data.

Reviewer #3: Study design

• indicate the rationale for including 5 villages from Afar and 1 village from Gambela region, and give prevalence estimates for the two regions

• indicate what data were collected (e.g. gender, age, village…)

• describe how sample were randomly assigned to pools of 5

• indicate whether pools were prepared from samples of the same village or whether village of collection was not considered in preparing pools

• the time to perform urine reagent strip tests was not considered in time-to-result analysis, the justification to not do so is not clear since 2 minutes per sample is a significantly relevant time 

• indicate how follow-up samples were employed

Diagnosis of S. haematobium

• this section is entirely missing and should be integrated in the text

• no description of urine filtration microscopy nor urine reagent strip is given

• no details are given on the qualification of the personnel performing tests in the field (individual testing) and in the laboratory (pooled testing) and how it might have affected the results

Statistical data analysis

• indicate that URS prevalence was also determined

• lines 155-164, since the distribution of infection intensity (i.e. eggs count) is not normal, and non parametric tests are used for comparisons between groups, the median should be computed and reported, not the mean

• lines 165-172, expand the description of the reference standard to make it clearer (e.g. IT+ and/or CPT+ was considered positive, IT- AND CPT- was considered negative)

• lines 173-180, clarify/simplify the description of the infection intensity (e.g. the infection intensity of the pool was estimated as the mean of infection intensity of the individual samples, the distribution of infection intensities was computed and quartiles identified, the prevalence of infection was then analysed according to levels i.e. quartiles)

• line 175, “intensitity” should be “intensity levels”

• line 185, “processing samples and” should be deleted (otherwise the following is not clear)

**Results**

-Does the analysis presented match the analysis plan?

-Are the results clearly and completely presented?

-Are the figures (Tables, Images) of sufficient quality for clarity?

Reviewer #1: (No Response)

Reviewer #2: The analysis is well outlined, executed and presented. Though it is somewhat limited and I have made some recommendations for further analysis of this valuable data set.

Reviewer #3: • Table 1: include 95% CI of UFM prevalence and UFM median egg count; p-values should be columns not rows; indicate the reference group for comparison; include URS prevalence, 95% CI and p-value

• Table 2: it would be preferable to swap columns (testing strategy) and rows (diagnostic test) so that results can be easily compared; the difference should be included before its 95% CI

• Figure 1: in the x axis, simply indicate quartile/level 1,2,3,4; use more differentiated colours for IT and CPT

• A justification should be given for interpreting r=0.57 as moderate correlation; an explanation should be given for the peculiar shape of the scatter plots with many data points flattened on the x or y axis

• Line 256, “when 10 ml and 15 ml…”, adding “both” would make the sentence clearer

• Time to results: it would be advisable to include a table with the calculations

**Conclusions**

-Are the conclusions supported by the data presented?

-Are the limitations of analysis clearly described?

-Do the authors discuss how these data can be helpful to advance our understanding of the topic under study?

-Is public health relevance addressed?

Reviewer #1: (No Response)

Reviewer #2: The conclusions of the study are supported by the data presented, however further investigations and conclusions regarding the usage of URS should be added. The conclusions do not address this test method sufficiently. The public health relevance is sufficiently addressed.

A second limitation relates to the testing of the pooled samples, it seems to me that only testing 10 or 15ml of the pooled samples is a severe limitation as the contribution of the individual samples is at this point severely diluted. In my opinion, this must be stated as a limitation, with some speculation about the utility or lack thereof of repeated testing for example.

Another point of interest would be to factor in the time needed to collect sufficient samples for pooling, would this lead to an overall delay in the processing of the samples? How were the samples stored in this case?

Line 325: the limitations regarding sample size from individual villages should be described in more detail to highlight the specific instances where this was a limitation and why.

Reviewer #3: • Lines 288-289, the concept should be expanded and clarified: cascade pooled testing can or cannot be used for planning and evaluating mass treatment programs? if yes, how?

• Lines 291-301, lower prevalence by CPT vs IT is explained with lower sensitivity at lower infection intensities. Because of a dilution effect? Should expand on that

• Lines 291-301, if lower sensitivity at lower infection intensity, how a similar infection intensity estimate can be explained? 

• Lines 297-301, please expand the very interesting consideration on pool sizes

• Lines 308-309, it would be advisable to change in “pooled testing my help better map the distribution of high intensity infections”

• Lines 325-6, power calculations were not shown

• Lines 330-1, it was not clear from study design that different samples contribute different volumes of urine to the pools, clarify that point and its implications

• A justification should be given for not conducting a cost-effectiveness analysis

**Editorial and Data Presentation Modifications?**

Reviewer #1: Some specific comments:

Abstract:

Why 10 and 15ml? Either the authors should explain why different volumes were used, or leave it out of the abstract.

Introduction

Line 94: the term ID has not been introduced before, please include.

Methods

Line 122: why adding formalin? 

Line 142-143: why UFM on both 10 and 15 mL in the CPT approach? Please include explanation here. 

Results

Table 1 mean UEC between regions, 2.78 in Afar and 3.05 in Gambella, the difference in mean UEC is quit small, how can this be statistically significantly different? Please check.

Overall, mean egg counts are relatively low, have the authors considered including the geometric mean or median for the egg count?

Line 239: the percentage for 15 mL should be 48.6% instead of 49.0%?

Discussion

Line 278-280, so no difference between 10 and 15mL? Also, indeed using more sensitive methods might improve the performance of pooled testing, not only molecular methods but also antigen detection methods could be highly useful (eg circulating anodic antigen).

How about those identified as positive by CPT but were initially negative by individual testing?

Line 288-289: so basically, the current CPT approach would not be recommended for planning and evaluating the success of MDA programs?

Line 291-292: I find it surprising that the mean UEC is the same between IC and CPT given the fact that the prevalence based on CPT was 50% lower compared to IT so more individuals negative based on CPT. What was the range in egg counts between the methods? Have the authors considered showing the geometric mean or the median egg count? Please clarify.

Line 298-299: UFM is less sensitive in detecting light intensity S. haematobium infection [19]. This is repetition, it is already mentioned in the previous paragraph, and in my opinion does not need to be repeated here.

Line 322-324: unclear what is meant by ‘… pooled urine samples of volumes 10 and 15 pooled samples.’ Please clarify.

Reviewer #2: No modifications of existing data recommended. 

In my opinion the abstract lacks clarity, particularly the ‘Methodology/Principal findings’ section. I suggest that this section is rewritten/restructured, to more clearly highlight the differences observed between the different test approaches, including segregating further the assessment of the test approaches, volume tested and time taken, as these all feature in a manner that is not immediately clear to the reader i.e. line 33 onwards would benefit from separation with a phrase such as – ‘We also compared the sensitivity of the test methods when testing 10ml vs 15ml of urine’ concluding that ‘we found this to be comparable’.

The conclusion states that the performance of pooled testing may be compromised in endemic areas with low intensity infection, however the reason for this is not clear from the abstract.

In general, from the abstract and author summary it is not clear which test approach performed better and why.

In the background section, lines 76-78, a description of current testing approaches is needed to provide the reader with more context to the problems faced.

Line 79: ‘needed’ instead of ‘most welcome’

Line 85: ‘imperfect and resource-demanding’ requires explanations as to why

Line 94: ‘ID’ should be ‘IT’

Line 139: delete ‘pool’ at end of line (duplicate)

Line 146: illustrates – spelling mistake

Line 195: ‘included’ instead of ‘involved’

Line 224: Bracket notation inconsistent as compared to lines 221 and 222

Table 2: Inclusion of raw numbers as in Table 1 would be helpful.

Fig 2: A description of the values that define the quartile ranges would be very helpful.

Line 239: 49% should be 48.6%

Line 253: CTA is not described

Reviewer #3: Abstract

• Line 25, UEC acronym should be introduced correctly as Urine Eggs Count, not urine samples

Background

• It would be advisable to expand the references to include more journal articles other than WHO reports

**Summary and General Comments**

Reviewer #1: The authors present a very interesting (follow-up) study in which they determine the clinical sensitivity and time-to-result of a pooled testing approach for assessing the prevalence and intensity of Schistosoma haematobium infections. A pooling approach sounds very promising, in particular given the advantage of saving time. However, even though the pooling method is substantially faster than testing all samples individually, the overall sensitivity of the pooling method is poor, which – in my opinion – is more important than saving time. Indeed, performance can be reduced when it comes to low intensity infections as microscopy in general has limited sensitivity in such cases. However, making pools of individual urines could also have a dilution effect and further reduce the sensitivity as it might be very difficult to detect only few eggs. The authors suggest to use higher volumes of the final pools for microscopy in order to improve accuracy, but would this be feasible? How would this reflect in time-to-result?

What is the added value of including urine reagent strips (URS)? It might be useful for screening but will never confirm Schistosoma infection. Also, with all the different abbreviations (UFM/URS/UEC/CPT/IT) I find it rather confusing to read and understand what the authors are presenting. 

One thing the authors have investigated is the difference in sensitivity based on the number of eggs counted in the individual testing. This sounds indeed logical, but this is something that you wouldn’t be able to control in real life settings if you do a pooled testing approach, so why present results as such?

Reviewer #2: Degarege et al. have a presented an interesting study on an important topic. The question of appropriate diagnostic approaches is an essential one in the fight to eliminate disease such as schistosomiasis and it is commendable that they pursue this question in such detail. Additionally, they have a wealth of data and a very valuable cohort at their disposal. It is for this reason that I recommend a deeper analysis of the data to provide a more valuable resource for the community and to maximize the potential of the data at their disposal.

The current analyses are sound and the methods and approaches taken are all of good rationale. They provide an interesting insight into the differences between individualized and pooled testing; highlighting important considerations such as time taken, in order to recommend the best approach going forward.

However, what is lacking in my opinion is a true and distinct analysis and recommendation. For example, the time taken to conduct individualized testing is ~1.5x more than pooled testing. Though there is the assumption that individualized URS testing is faster than either approach and with a sensitivity of 68.8%, easily more sensitive than pooled UFM testing. Therefore, it seems that this would be the prudent approach; further comment is needed here. And importantly further analysis, as there is no commentary on the specificity of URS. To be able to recommend URS as faster screening approach (which seems the obvious outcome here) an analysis of specificity in this cohort must be conducted. This will add considerable value to the study. Some more information on the methodology used for the URS would then also be helpful.

This is important in order to provide further novelty, which is highlighted as a reason for pursuing this study after the authors’ initial 2015 study. URS is cited as a novelty factor in the current study but in my opinion needs to be investigated further to substantiate this claim.

Finally, in my opinion, some additional post-hoc analysis should be conducted regarding pool sizes. In the discussion it is stated that preferred pool sizes need further investigation, so my question is why not conduct this investigation as part of this paper? This would add considerable value here. The authors currently have a varied number of pool sizes with existing data, I would encourage them to conduct a sub-analysis here if possible – this may not be possible since as I understand smaller pool sizes were only tested in case of positivity, but I encourage the authors to revisit this data nonetheless. If this is not possible, I would encourage to at least have a detailed breakdown of the positive pools, as I believe this would still be informative. For example, starting at the largest positive pool size and describing within this pool how many sub-pools were positive, until the individual positive samples are identified. This is also true for the egg counts found in each pool. These data can be included as supplementary data.

Reviewer #3: The manuscript by Degarage and colleagues aims at comparing Schistosoma haematobium cascade pooling testing vs individual testing in terms of i) prevalence of infection ii) intensity of infection iii) time-to-result. The study was conducted among school-aged children in villages from Afar and Gambella regions in Ethiopia, and employed parasitological (urine filtration microscopy) and lateral-flow (urine reagent strips) assays for diagnosis. The results are of relevance for the epidemiology and control of S. haematobium infection in Sub Saharan Africa: while cascade pooled testing lacks the necessary sensitivity needed for estimating population infection prevalence, it might be a faster method to detect transmission hotspots and estimate population infection intensity. However, the originality is limited since both the authors and others have previously provided similar evidence, cost-effective analysis is not conducted, and real life cases for applications are not discussed. Also, methodology lacks some important details, and reporting and discussion of the results needs to be improved for better clarity.

PLOS authors have the option to publish the peer review history of their article (what does this mean?). If published, this will include your full peer review and any attached files.

Reviewer #1: Yes: P.T. Hoekstra

Reviewer #2: No

Reviewer #3: No
---

## [Decision Letter · Decision Letter 1]

5 Jul 2024

Dear Dr Degarege,

Thank you very much for submitting your manuscript "Clinical sensitivity and time-to-result of a cascaded pooled testing approach for assessing the prevalence and intensity of Schistosoma haematobium infections" for consideration at PLOS Neglected Tropical Diseases. As with all papers reviewed by the journal, your manuscript was reviewed by members of the editorial board and by several independent reviewers. In light of the reviews (below this email), we would like to invite the resubmission of a significantly-revised version that takes into account the reviewers' comments. 

While the manuscript has clearly improved according to the reviewers' comments, still some substantial adaptations are required to allow acceptance.

We cannot make any decision about publication until we have seen the revised manuscript and your response to the reviewers' comments. Your revised manuscript is also likely to be sent to reviewers for further evaluation.

Sincerely,

Daniela Fusco, PhD

Academic Editor

Jong-Yil Chai

Section Editor

While the manuscript has clearly improved according to the reviewers' comments, still some substantial adaptations are required to allow final acceptance.

Reviewer's Responses to Questions

**Key Review Criteria Required for Acceptance?**

**Methods**

-Are the objectives of the study clearly articulated with a clear testable hypothesis stated?

-Is the study design appropriate to address the stated objectives?

-Is the population clearly described and appropriate for the hypothesis being tested?

-Is the sample size sufficient to ensure adequate power to address the hypothesis being tested?

-Were correct statistical analysis used to support conclusions?

-Are there concerns about ethical or regulatory requirements being met?

Reviewer #1: (No Response)

Reviewer #2: One limitation is the lack of a specific reference test e.g. PCR testing for example. The gold standard used here is a composite of IT and CPT, however I think it is a limitation that there is no independent test method used as a gold standard and this can be stated as such.

I also suggest that a more comprehensive overview of existing test methods be included in the introduction around line 87.

Reviewer #3: The aim of the study is to investigate whether cascade pool testing might be a cost-effective alternative to individual testing for estimating the prevalence and intensity of S. haematobium at population level, by comparing CPT and IT performed by urine filtration microscopy or urine reagent strip, and by using 10ml or 15ml of urine volume.

The study design is not entirely appropriate to address the aim and objectives since time-to-result analysis is limited and cost-effectiveness analysis was not conducted.

Power analysis was not conducted and the sample size was considered insufficient to conduct the analysis at village level.

Statistical methods should be carefully revised. Variation of prevalence and intensity of S.haematobium infection with demographic factors should be evaluated by multivariate analysis. Egg counts should be analysed by non-parametric tests. Egg counts should be analysed among positive samples only.Other specific suggestions are given below.

**Results**

-Does the analysis presented match the analysis plan?

-Are the results clearly and completely presented?

-Are the figures (Tables, Images) of sufficient quality for clarity?

Reviewer #1: (No Response)

Reviewer #2: The authors made an effort to include further analysis and the inclusion of Table 3 is noted. However more description of this table in the results section is required. Particularly explaining the rationale as to why not all positive pools were then tested.

Furthermore, specificity values as referred to in line 374 should be conducted or included as a Table/Figure. Including at the minimum specificity values but possibly also negative and positive predictive values would be helpful.

Reviewer #3: The results are still not clearly presented. The tables needs careful revision and integration. The quality of the immages should also be improved. Specific suggestions are given below.

**Conclusions**

-Are the conclusions supported by the data presented?

-Are the limitations of analysis clearly described?

-Do the authors discuss how these data can be helpful to advance our understanding of the topic under study?

-Is public health relevance addressed?

Reviewer #1: (No Response)

Reviewer #2: The authors provide a solid overview of the limitations of this method, it seems clear that the sensitivity of such an approach is lacking, however it is a topic which merits investigation and so it is beneficial to know the best approaches to explore and those to avoid.

Line 329 - An alternative argument is that IT UFM suffers from a lack of sensitivity, which is true particularly when testing samples once. Here this could instead argue for repeated testing of the same samples or participants.

Reviewer #3: The authors discuss strenghts and limitations of their results in view of the literature. However, the significance of the study and the advancement in the topic under study are limited by issues with methodology and results presentation.

**Editorial and Data Presentation Modifications?**

Reviewer #1: (No Response)

Reviewer #2: Line 339 - include figure reference

Spelling and grammar mistakes e.g. line 373 - should be 'diagnosis of'

Reviewer #3: • It is still not clear why cost-effectiveness was not evaluated despite its importance is stressed in the introduction (“However, these surveys are resource demanding, and more cost-effective strategies are needed. A cost-saving alternative is a pooled sample testing approach” and “it remains unclear whether such a CPT would reduce the time-to-result, and therefore can be recommended as a cost-efficient alternative to the current diagnostic approach”), despite such evaluation is stated as one aim of the study in the methods (“To verify whether our CPT is a cost-saving strategy”) and despite it could be estimated even retrospectively

• Is it still not clear why time-to-result analysis includes UFM but not URS performed on pools vs individual spercimens, this could be done even retrospectively since URS should take a standard amount of time per sample

• It is still not clear how data were collected: there is no mention of data collection forms and data management procedures

• It is still not clear how the samples were randomised for grouping and pooling: there is no mention of randomization procedures

• Why parametric tests were used to compare the distribution of eggs number between CPT and IT, given the distribution is not normal? In the previous version of the manuscript, non parametric tests were correctly used. In the point-to-point answers to reviewers comments, it is stated that parametric tests can be used since the sample size is large (?), but no justification is given for not using non-parametric tests. Also, in some part of the manuscript a geometric mean is reported, in others the arithmetic mean. I reckon that boxplot should be used instead of bars to show distribution of egg counts and that non-parametric tests should be used for comparison. Also, the comparison in distribution of egg counts, as well as their correlation, should be tested among positive samples only. The results and interpretation have changed from the previous version of the manuscript. I strongly suggest a thourough statistical revision of the manuscript.

• The comparison of prevalence and intensity of S. haematobium infection according to demographic factors (Table 1) should be performed by multivariate analysis 

• The title of the first results paragraph and the title of Table 1 should be “Prevalence and intensity of…”

• Table 1 column 3 should be “Percentage of IT/CPT positive samples by UFM (95% CI)”

• Table 1 column 4 should be “Egg count (geometric mean) by UFM”

• Table 1 column 5 should be “Percentage of IT/CPT positive samples by URS (95 % CI)”

• In Table 1 and Table 2 the 95% CI of the proportion would be better indicated by the lower confidence limit – upper confidence limit (use dash instead of comma)

• In Table 1, p-values would be better reported in a column not a row, since it is not a category of the variables (this was already suggested in the first review)

• In Table 2 column 1, Urine Filtration Microscopy should be used instead of microscope

• In Table 2 column 5, change title to “Difference between IT and CPT when using 10 ml samples (95% CI)

• In Table 2, the p-value for the difference should be reported

• “UFM was 100% specific in identifying urine samples without S. haematobium egg through the IT and CPT approach”: this was an assumption on which the standard reference for comparing IT and CPT was built, therefore cannot be a result

• Table 3 is not at all clear and is not described neither by a legend or in the text. What analysis was conducted, to what aim? Why not all pools were tested? Why some pools were assumed to be negative?

• Figure 3: correlation between egg counts should be performed on positive samples only. The sensitivity, i.e. ability to detect positive samples, was already compared.

• Time-to-result data and results paragraph is extremely limited, and it seems insufficient given the importance of this aspect in the significance of the study

• “The scatter plot showed positive trends in the relationship of the mean UECs of the individual urine samples estimated based on the IT and the UECs estimated using the CPT. There was also a moderate correlation in UEC estimated by the IT and CPT approach” The two sentences are redundant, please rephrase.

• The clarity of the discussion could benefit from being restructred so that first results are summarised and interpreted, then limitations are discussed, and finally their significance is described in the light of the existing literature and of the study aims

**Summary and General Comments**

Reviewer #1: The authors have correctly addressed all my comments and questions and I am satisfied with the changes made in the manuscript. The revised manuscript is acceptable for publication.

Reviewer #2: The authors have made a comprehensive effort to address the comments on the initial manuscript, and I believe the current manuscript sufficiently highlights the limitations of this approach, while still indicating that this is an important topic that requires further study.

Reviewer #3: The overall clarity of the manuscript has improved in its revised version. However, some important methodological aspects were not addressed by the authors in their revision, and results presentation needs further revision to allow a correct interpretation. Because of these limitations, the conclusions of the study and its significance seem limited.

PLOS authors have the option to publish the peer review history of their article (what does this mean?). If published, this will include your full peer review and any attached files.

Reviewer #1: Yes: Pytsje T Hoekstra

Reviewer #2: No

Reviewer #3: No
---

## [Decision Letter · Decision Letter 2]

5 Aug 2024

Dear Dr Degarege,

We are pleased to inform you that your manuscript 'Clinical sensitivity and time-to-result of a cascaded pooled testing approach for assessing the prevalence and intensity of Schistosoma haematobium infection' has been provisionally accepted for publication in PLOS Neglected Tropical Diseases.

Best regards,

Daniela Fusco, PhD

Academic Editor

j

Jong-Yil Chai

Section Editor

Reviewer's Responses to Questions

**Key Review Criteria Required for Acceptance?**

**Methods**

-Are the objectives of the study clearly articulated with a clear testable hypothesis stated?

-Is the study design appropriate to address the stated objectives?

-Is the population clearly described and appropriate for the hypothesis being tested?

-Is the sample size sufficient to ensure adequate power to address the hypothesis being tested?

-Were correct statistical analysis used to support conclusions?

-Are there concerns about ethical or regulatory requirements being met?

Reviewer #2: (No Response)

**Results**

-Does the analysis presented match the analysis plan?

-Are the results clearly and completely presented?

-Are the figures (Tables, Images) of sufficient quality for clarity?

Reviewer #2: (No Response)

**Conclusions**

-Are the conclusions supported by the data presented?

-Are the limitations of analysis clearly described?

-Do the authors discuss how these data can be helpful to advance our understanding of the topic under study?

-Is public health relevance addressed?

Reviewer #2: (No Response)

**Editorial and Data Presentation Modifications?**

Reviewer #2: (No Response)

**Summary and General Comments**

Reviewer #2: (No Response)

PLOS authors have the option to publish the peer review history of their article (what does this mean?). If published, this will include your full peer review and any attached files.

Reviewer #2: No

---

## [Editor Report · Acceptance letter]

15 Aug 2024

Dear Dr Degarege,

We are delighted to inform you that your manuscript, "Clinical sensitivity and time-to-result of a cascaded pooled testing approach for assessing the prevalence and intensity of Schistosoma haematobium infection," has been formally accepted for publication in PLOS Neglected Tropical Diseases.

Best regards,

Shaden Kamhawi

co-Editor-in-Chief

Paul Brindley

co-Editor-in-Chief
